# Ammonia Influences the Zooplankton Assemblage and Beta Diversity Patterns in Complicated Urban River Ecosystems

**Caili Du** [1,2,3,†]**, Fengbin Zhao** [1,†]**, Guangxia Shang** [2,3]**, Liqing Wang** [2]**, Erik Jeppesen** [4,5,6]**, Lieyu Zhang** [3,*]**, Wei Zhang** [2,*] **and Xin Fang** [7]



1   State Key Laboratory of Pollution Control and Resource Reuse, College of Environmental Science and Engineering, Tongji University, 1239 Siping Road, Shanghai 200092, China; aquaducl@163.com (C.D.); fbzhao@tongji.edu.cn (F.Z.)
2   Engineering Research Center of Environmental DNA and Ecological Water Health Assessment, Shanghai Ocean University, Shanghai 201306, China; aquashanggx@163.com (G.S.); lqwang@shou.edu.cn (L.W.)
3   State Key Laboratory of Environmental Criteria and Risk Assessment, Chinese Research Academy of Environmental Sciences, Beijing 100012, China
4   Department of Ecoscience, Aarhus University, 6000 Aarhus, Denmark; ej@ecos.au.dk
5   Sino-Danish Centre for Education and Research (SDC), University of Chinese Academy of Sciences, Beijing 100190, China
6   Limnology Laboratory, Department of Biological Sciences and Centre for Ecosystem Research and Implementation, Middle East Technical University, Ankara 06800, Turkey
7   School of Business, Macau University of Science and Technology, Macao 999078, China
*   Correspondence: zhangly@craes.org.cn (L.Z.); weizhang@shou.edu.cn (W.Z.)
†   These authors contributed equally to this work.

**Abstract:** Beta diversity represents the spatial or temporal variation of species diversity among sampling sites and may be composed of two elements: turnover (Brepl, replacement of species assemblages) and nestedness (Brich, loss or gain of species assemblages). Knowledge of the mechanisms driving beta diversity contributes to an understanding of the variation in aquatic ecosystem community structures. We sampled zooplankton assemblages at 24 sites in 11 rivers in Shanghai City and conducted a nutrient addition experiment to elucidate the effects of various environmental variables on the community structure and beta diversity patterns of the zooplankton. The zooplankton assemblages in the rivers differed significantly at ammonia nitrogen ($NH_3$-N) concentrations below (Group I) and above (Group II) 1.03 mg·$L^{-1}$. The nutrient addition experiment further demonstrated that the composition of the zooplankton assemblages changed markedly along an ammonia concentration gradient (0.2 to 5.0 mg N·$L^{-1}$). The total beta diversity of Group I was considerably higher than that of Group II, indicating that high nutrient (ammonia) pollution led to biotic homogeneity. Overall, turnover was the key factor determining the total beta diversity of the two groups, suggesting the key importance of replacement of zooplankton assemblages. Furthermore, we found a correlation between environmental factors (mainly nitrogen content) and the beta diversity of zooplankton, and beta diversity (Brepl and Btotal) decreased with increasing trophic state. These findings provide further insight into the changing characteristics of the beta diversity patterns of zooplankton in river networks and may help to guide managers dealing with conservation strategies for aquatic biodiversity preservation in urban river ecosystems.

**Keywords:** zooplankton; community structure; beta diversity; turnover; nestedness; river ecosystem





## 1. Introduction

Biodiversity, typically including alpha, beta, and gamma components, is of key importance for maintaining basic ecological functions and a stable and healthy environment in an ecosystem [1]. Beta diversity describes the spatial or temporal variation of community composition among sites and consists of two elements: turnover (replacement of species

assemblages) and nestedness (loss or gain of species assemblages) [2]. These two components jointly affect the compositions of communities and their degree of heterogeneity [3–6], and reflect the mechanisms involved in community structure and diversity patterns in the spatial dimension [7,8]. The turnover component indicates the replacement or substitution of some species by others between assemblages. In contrast, nestedness reflects the loss or gain of species composition, in which sites with a poor richness of species are a strict subset of another site with higher species richness [9,10]. In general, environmental filters, competition, and historical events all contribute to species replacement mechanisms [11]; nestedness, on the other hand, occurs through species thinning or other ecological processes, such as selective extinction, colonization, or dispersal limitation, which results in species-poor sites that are a subset of the region's richest site [10]. Moreover, differences in the environment, organisms, and ecosystems can lead to changes in the importance of the two components in the pattern of beta diversity [11–13], with implications for conservation strategies aiming to protect biodiversity in these ecosystems.

The zooplankton community occupies an important position in an aquatic ecosystem and serves as a transitional part of the food chain, as these organisms not only consume primary producers (phytoplankton) by grazing but also act as food for higher trophic levels (fish) [14,15]. Moreover, different species of zooplankton are sensitive to environmental changes to various degrees, thus making zooplankton a good indicator of environmental quality [16–18]. Several studies have shown that environmental changes (nutrient enrichment) have a marked impact on the growth, community composition, density, and diversity of zooplankton in aquatic ecosystems [19–21]. Recently, researchers have shown an increased interest in variation in community composition among sites (beta diversity), which is also considered a very practical tool for exploring the effects of environmental factors on zooplankton communities [22,23]. Data from several studies suggest that the construction of dams or flooding of river–floodplain systems may induce environmental changes, which in turn could alter zooplankton community structures and decrease zooplankton beta diversity [24,25]. Rivers are important ecosystems for the development of urban centers [26]. Rapid urbanization and rising living standards have, however, resulted in the release of large amounts of industrial and domestic wastewater into rivers, which has exacerbated water pollution and environmental deterioration due to high nutrient loads [27–29]. More and more rivers in urban areas are facing eutrophication (nutrient enrichment) due to increasing anthropogenic activities [30,31], which has led to obvious changes in species composition and diversity. Additionally, the study of the beta diversity patterns and community structures of zooplankton in complicated urban river ecosystems contributes to an understanding of the overall state of these ecosystems and may provide guidance for river ecosystem restoration and conservation.

In this study, we used rivers in Shanghai City, China, constituting a complex urban river network, to investigate the in situ effects of key environmental factors on zooplankton assemblages and beta diversity distribution patterns. Furthermore, we conducted an indoor nutrient addition experiment to verify the impacts. We hypothesized that (1) environmental factors would be the key factors affecting the structure and beta diversity of zooplankton assemblages; (2) nutrient enrichment would lead to the reduction of beta diversity of zooplankton in complicated urban river ecosystems; (3) the turnover component (replacement of species, Brepl) would play a leading role in the beta diversity patterns of zooplankton assemblages in habitats highly disturbed by anthropogenic activities, given that it is commonly discovered to be the dominant component. Our observations provide new insight to support the evaluation of the correlation between nutrient status and the community structure and beta diversity pattern of zooplankton in urban river ecosystems; this is also of relevance for ecosystem restoration and conservation strategies.

## 2. Materials

### 2.1. Study Region

A total of 24 sampling points in 11 study rivers located in four districts of Shanghai City were investigated in April and July 2017 (Figure 1, Table S1). The study area experiences a subtropical humid monsoon climate, with an average temperature of 17.7 °C, an average annual precipitation of 1388.8 mm, and an average annual evaporation of 1111.6 mm in 2017. The rainy season generally occurs from June to September, accounting for more than 60% of the annual precipitation. Of the sampling points, 13 (S1–S5, S8–S10, and S20–S24) are surrounded by densely populated residential areas, and the river banks have been fixed (concrete or stone). The other sampling sites are surrounded by agricultural land, and the river banks are in a natural state (S6, S7, and S11–S19). The rivers have received excessive nutrient loading due to rapid industrialization and urbanization, and nonpoint pollution has led to eutrophication of these rivers as demonstrated by, for instance, cyanobacterial blooming in summer.

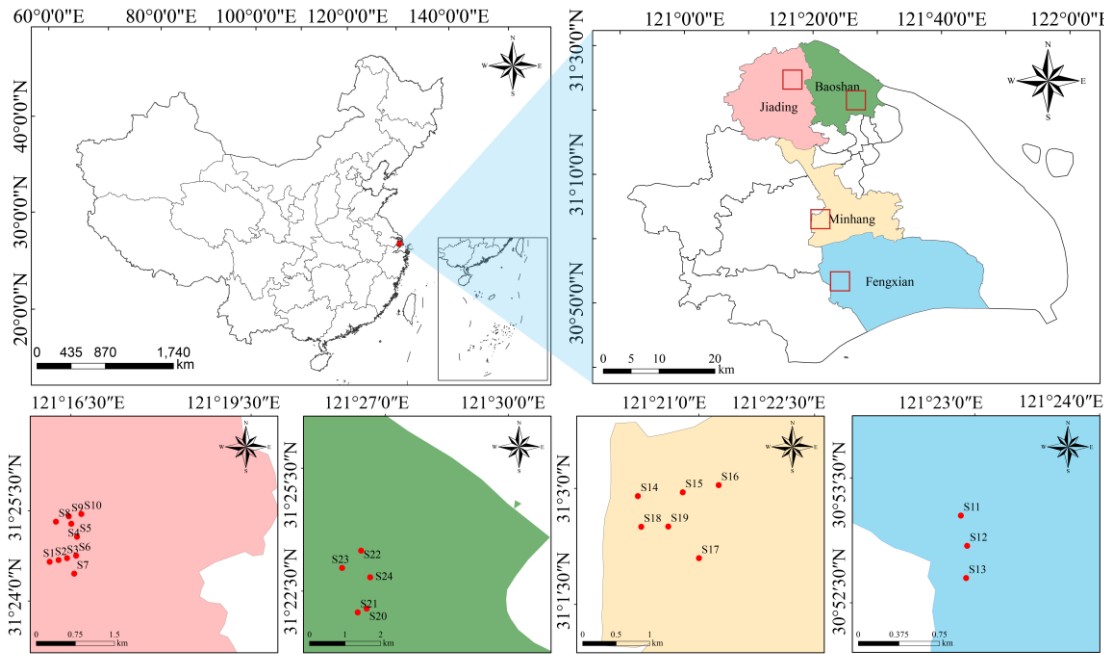

**Figure 1.** Sampling sites in the rivers of Shanghai City.

### 2.2. Sample Collection and Laboratory Analyses

Environmental factors including water temperature (WT), pH, conductivity (Cond), and dissolved oxygen (DO) were tested on-site using a YSI Pro Plus at a fixed depth (0.7 m). The water samples (n = 48) were taken at depths of approximately 0.5 m and 1 m below the water surface using a large plastic container at each sampling site, together with 500 mL mixed water samples, and the samples were analyzed no later than 24 h after sampling. Chlorophyll a (Chla) was measured using the acetone extraction method, and total nitrogen (TN), ammonia nitrogen ($NH_3$-N), nitrate nitrogen ($NO_3$-N), nitrite nitrogen ($NO_2$-N), total phosphorus (TP), total dissolved phosphorus (TDP), and chemical oxygen demand (COD) were analyzed in the laboratory following standard methods [32].

The zooplankton (n = 48) were divided into three groups: Rotifera, Copepoda, and Cladocera. For Rotifera determination, a 1 L mixed water sample was taken from the surface (at depths of approximately 0.5 m and 1.0 m below the surface) at each sampling site; the sample was preserved with 4% formaldehyde in the field and then, after sedimentation for 48 h, concentrated to a final volume of 30 mL for quantitative analysis. Counting of Rotifera in 1 mL measures of the concentrated sample was conducted at 10 × 10 magnification using a microscope, and the Rotifera were identified to the species or genus level [33].

Each sample was counted three times. For the crustacean zooplankton (Cladocera and Copepoda), 20 L mixed water samples were collected from the shore in the daytime between 9th April and 29th July and then filtered on a 64 μm plankton net, preserved in 4% formaldehyde, and concentrated to a final volume of 25 mL after sedimentation for 48 h. The fully concentrated samples from each site were used for counting and identification under a microscope [33,34] at $10 \times 10$ magnification. Most taxa were identified as species. Adult copepods were identified to the species level. Larval and juvenile forms were classified as nauplii or Cyclopoid copepodites (Calanoida and Harpacticoida were rarely found in this study) and were excluded from the analyses concerning species.

### 2.3. Experimental Design

As ammonia turned out to be of key importance in the field, we conducted a laboratory experiment with seven different concentrations of ammonia. A total of 28 cylindrical, open-bottomed, high-density polyethylene mesocosms (100 cm in height, 50 cm diameter) were immersed in 40 L lake water (taken from Minghu Lake, which is connected to Dishui Lake, on the campus of Shanghai Ocean University). The concentration of $NH_3$-N in Minghu Lake was $0.18 \pm 0.01$ mg $N \cdot L^{-1}$, that of TN was $0.65 \pm 0.04$ mg $\cdot L^{-1}$, and that of TP was $0.13 \pm 0.01$ mg $\cdot L^{-1}$. $NH_3$-N was added (using $NH_4Cl$) to produce final concentrations of 0.6 mg $N \cdot L^{-1}$ to 5.0 mg $N \cdot L^{-1}$ [35] (Table 1, Figure S1), and each treatment had three replicates. During the experimental period of 10 days, WT fluctuated between 20 °C and 25 °C, and pH was slightly alkaline with pH > 8. On day 10, each mesocosm was filtered through a 64 μm plankton net for analysis of zooplankton.

**Table 1.** Ammonia nitrogen concentrations in the different treatments (mg $\cdot L^{-1}$).

| Treatment Groups | A (Initial Value) | B | C | D | E | F | G |
|---|---|---|---|---|---|---|---|
| $NH_3$-N concentration (mg $N \cdot L^{-1}$) | 0.2 | 0.6 | 1.0 | 1.8 | 2.6 | 3.4 | 5.0 |

### 2.4. Analytical Methods

We used multiple regression tree (MRT) analysis to identify the potential drivers influencing the spatial variation of the zooplankton community structure [36] using the R package mvpart [37]. Principal coordinate analysis (PcoA) was conducted using a Bray–Curtis similarity matrix generated from square-root-converted zooplankton abundance data using the function betadisper in the R package vegan [38]. Permutational multivariate analysis of variance (PERMANOVA) (999 permutations) was used to analyze significant variations in zooplankton community composition and beta diversity indices between the groups. Analysis of variance (ANOVA) was used to estimate significant variations in the abundance of zooplankton among the groups in the experimental conditions (the Shapiro–Wilk test and Bartlett's tests were conducted to analyze the normality and homogeneity of raw data before ANOVA). The Wilcoxon rank-sum test was applied to evaluate significant variations in environmental variables and abundances of zooplankton between the groups in the urban river ecosystems (false discovery rate adjusted $p < 0.05$). The indicator species in each group was calculated using the function indval in the R package labdsv. Taxa with $p < 0.05$ were designated as indicator species. The dominant species were identified as the species with Y > 0.02 [39].

Total beta diversity (Btotal) of the zooplankton assemblages was calculated as described by Baselga (2010) and Podani and Schmera (2011) [2,40] and further partitioned into turnover (Brepl) and nestedness (Brich) (Btotal = Brepl + Brich), where (1) Btotal was based on the Sorensen dissimilarity index and represented the total variation within the community. (2) Brepl was quantified from the Sorensen dissimilarity index, reflecting species replacement. (3) Brich was the nestedness-resultant fraction of Sorensen dissimilarity, reflecting species loss/gain (richness differences) alone. The beta diversity indices (average values derived from the dissimilarity matrices) were calculated using the function beta.multi in the R package BAT. Moreover, three distance matrices between sites were

obtained by calculating abundance data for the zooplankton community structure using the function beta in the R package BAT. Beta diversity (average distance to the centroid) was calculated at each sampling site based on the aforementioned three matrices using the function betadisper in the R package vegan. Linear regression and the Mantel test (999 permutations) were applied to analyze the relationships between beta diversity (Btotal, Brepl, and Brich) and the content of $NH_3$-N using the R package vegan [38].

## 3. Results

### 3.1. Zooplankton Assemblages in Urban Rivers of Shanghai City

In the MRT, the zooplankton community structure created from square-root-transformed zooplankton abundance data at the sites was used as the independent variable, and environmental variables included WT, pH, DO, TN, Cond, TP, $NH_3$-N, Cond, and COD. The MRT divided the samples into two communities (Group I and Group II) with 1.03 mg·L$^{-1}$ of ammonia nitrogen concentration as a split node based on the "1-SE" principle (Figure 2).

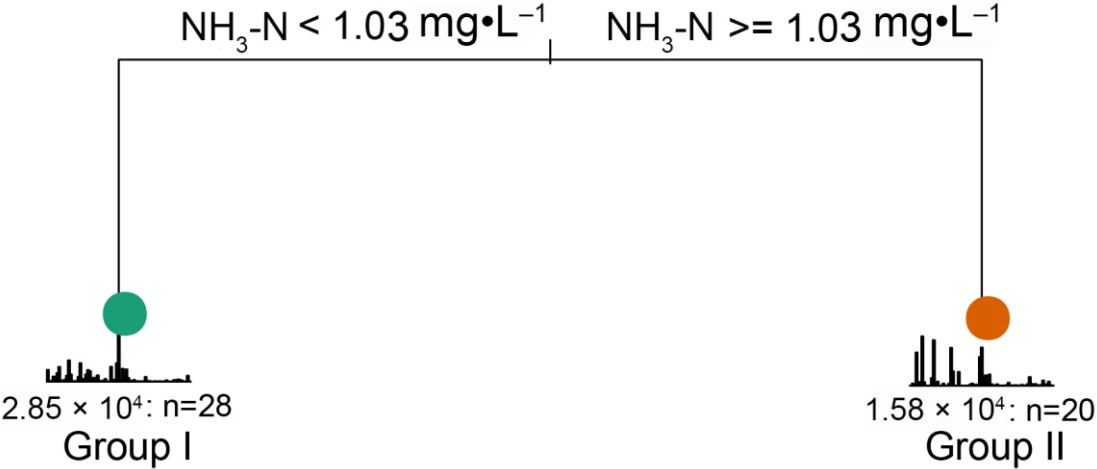

**Figure 2.** Dendrogram of multivariate regression tree classification based on zooplankton community structure in the rivers of Shanghai City.

A total of 75 zooplankton species (39 Rotifera, 19 Copepoda, and 17 Cladocera) were identified, of which 65 occurred in Group I and 50 in Group II (Table S2). The zooplankton density of Group I ranged from 54 to 8069 ind. L$^{-1}$ (average 1959 ind. L$^{-1}$) and that of Group II ranged from 709 to 5201 ind. L$^{-1}$ (average 2133 ind. L$^{-1}$) (Figure 3a). Among the three taxonomic groups, Rotifera dominated, accounting for 95.6% and 96.0% of the abundance in Group I and Group II, respectively, while the relative abundance of Cladocera was 2.0% in Group II and 0.8% in Group I (Figure 3b).

The two groups had different zooplankton indicator taxa, which in Group I ($NH_3$-N < 1.03 mg·L$^{-1}$) were the Rotifera *Polyarthra dolichoptera*, *Anuraeopsis fissa*, *Trichocerca pusilla*, *Keratella valga*, *Brachionus forficula*, and *B. falcatus*, the Cladocera *Diaphanosoma leuchtenbergianum,* and *Moina micruraKurz,* and the Copepoda *Mesocyclops leuckarti*, while in Group II ($NH_3$-N ≥ 1.03 mg·L$^{-1}$) the species were the Rotifera *Brachionus calyciflorus*, *Polyarthra trigla*, *Keratella cochlearis*, *K. quadrata*, and *Asplachna priodonta* and the Cladocera *Bosmina coregoni*, *B. longirostris*, *B. deitersi,* and *Chydorus sphaericus*. The relative abundances of the taxa differed markedly between the groups. *Polyarthra dolichoptera* was the numerically dominant species in Group I and *Brachionus calyciflorus* was the dominant species in Group II, i.e., the relative abundance of the rotifer *B. calyciflorus* was highest in Group II (20.9%) and lowest (3.7%) in Group I. The two groups had nine species in common (Table S2).

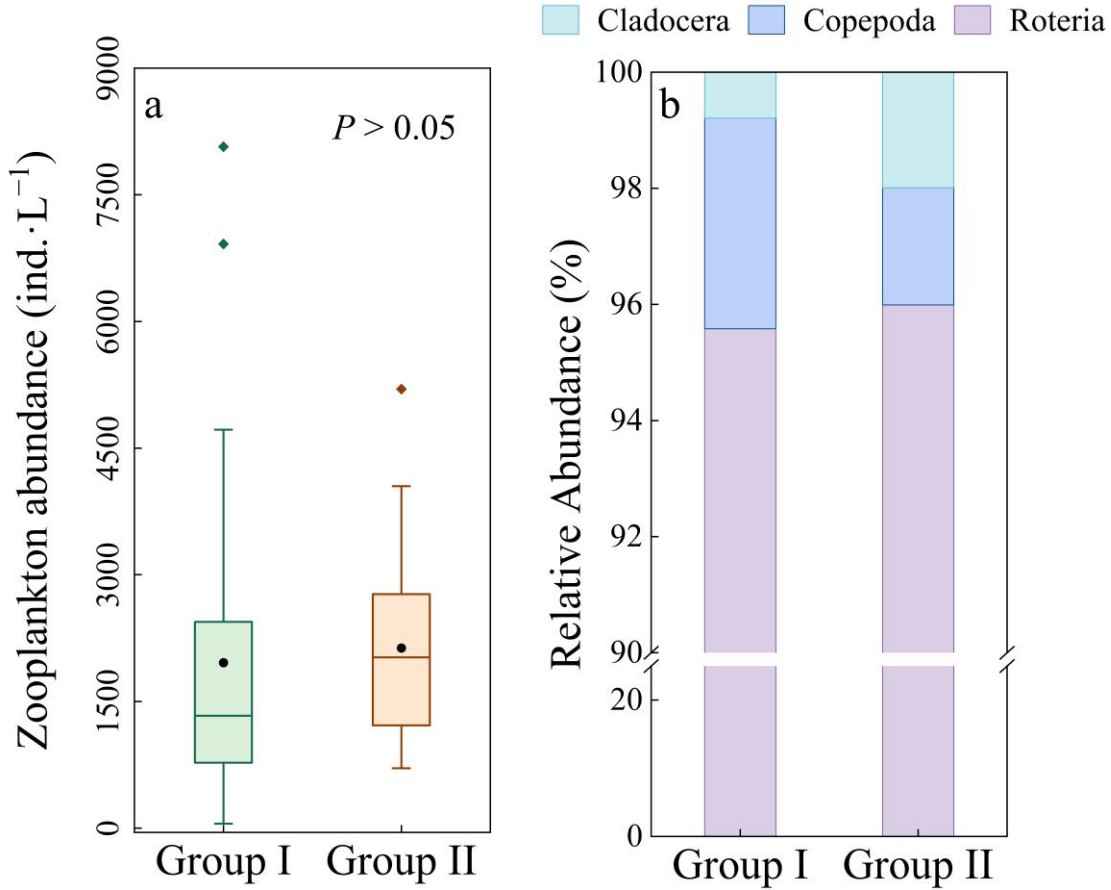

**Figure 3.** Zooplankton abundance (**a**) and relative abundance (**b**) in Group I (NH$_3$-N < 1.03 mg·L$^{-1}$) and Group II (NH$_3$-N $\geq$ 1.03 mg·L$^{-1}$).

The PcoA of the zooplankton abundance-based distributional matrix based on Bray–Curtis dissimilarity showed wide differences in zooplankton community structure between the groups (Figure 4a), and 38.3% of the variation was explained by PcoA1 and PcoA2. The result of the PERMANOVA with 999 permutations also demonstrated significant differences in the structure between the two groups ($p < 0.001$) (Figure 4b).

### 3.2. Environmental Variables

The environmental variables in the two groups of rivers are depicted in Figure S2. Some variables (COD, TN, Chla, NH$_3$-N, pH, DO, and WT) differed significantly between Group I and Group II ($p < 0.05$). The concentration of NH$_3$-N ranged from 0.2 to 1.0 mg N·L$^{-1}$ (average 0.6 mg N·L$^{-1}$) in Group I and from 1.0 to 4.5 mg·L$^{-1}$ (average 1.9 mg·L$^{-1}$) in Group II. COD, DO, pH, and TN in Group I were lower than in Group II, while the WT of Group I was higher than that of Group II. Chla ranged from 6.2 to 209 μg·L$^{-1}$ and 23.6 to 447 μg·L$^{-1}$ in Group I and Group II, respectively, with mean values of 55.3 mg·L$^{-1}$ and 125 mg·L$^{-1}$. Other variables, such as TP and Cond, did not differ between the two groups.

### 3.3. Beta Diversity of Zooplankton Assemblages

The total beta diversity of zooplankton had mean values of 0.56 and 0.41 in Group I and Group II, respectively. The turnover (Brepl) was the key component of total beta diversity for both groups, accounting for 65.4% and 60.5% in Group I and Group II, respectively. However, despite significant variations in Btotal and Brepl between the groups ($p < 0.05$), no significant differences were found for Brich ($p > 0.05$) (Table 2).

**Table 2.** The average and variance values of beta diversity in Group I and Group II.

| Beta Diversity | Group I (NH$_3$-N < 1.03 mg·L$^{-1}$) | Group II (NH$_3$-N ≥ 1.03 mg·L$^{-1}$) |
|---|---|---|
| Btotal * | 0.56 (0.04) | 0.41 (0.01) |
| Brepl * | 0.37 (0.03) | 0.24 (0.01) |
| Brich | 0.20 (0.01) | 0.16 (0.01) |

Note: * $p < 0.05$.

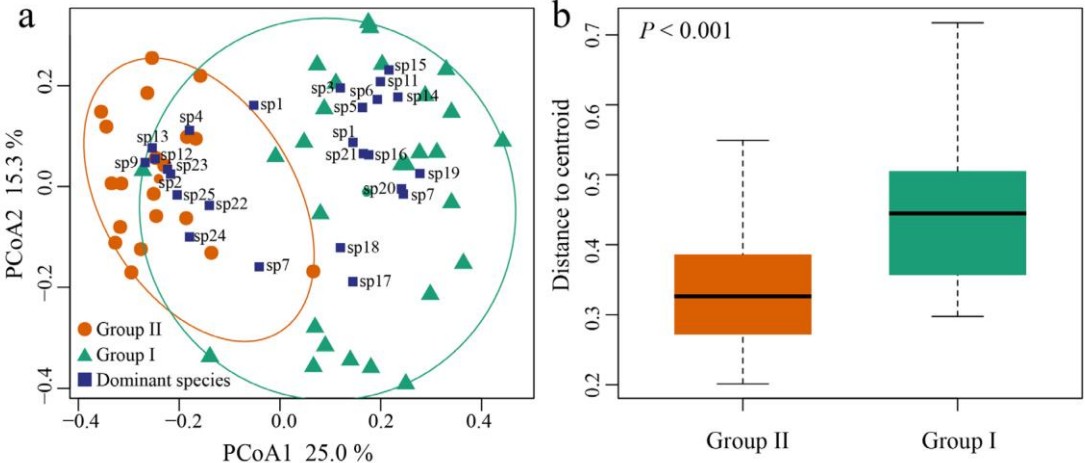

**Figure 4.** PcoA showed the differences in zooplankton community structure (**a**) and average distance to centroid (**b**) between Group I (NH$_3$-N < 1.03 mg·L$^{-1}$) and Group II (NH$_3$-N ≥ 1.03 mg·L$^{-1}$). sp1, *Polyarthra dolichoptera*; sp2, *P. trigla*; sp3, *Brachionus angularis*; sp4, *B. calyciflorus*; sp5, *B. forficula*; sp6, *B. falcatus*; sp7, *Keratella cochlearis*; sp8, *K. valga*; sp9, *K. quadrata*; sp10, *Filinia longisela*; sp11, *Anuraeopsis fissa*; sp12, *Asplanchna* sp.; sp13, *A. priodonta*; sp14, *Trichocercas* sp.; sp15, *T. pusilla*; sp16, *Thermocyclops taihokuensis*; sp17, *T. kawamurai*; sp18, *Mesocyclops leuckarti*; sp19, *Diaphanosoma leuchtenbergianum*; sp20, *Moina* sp.; sp21, *M. micrura*; sp22, *Bosmina coregoni*; sp23, *B. longirostris*; sp24, *Bosminopsis deitersi*; sp25, *Chydorus sphaericus*.

The linear model revealed close correlations between various environmental variables and the beta diversity of zooplankton assemblages (Table S3), with nitrogen variables (TN and NH$_3$-N) being of key importance for beta diversity (Btotal and Brepl), while Brich was not related to any environmental variable. Btotal was associated with NH$_3$-N ($R^2$ = 0.082, $p < 0.05$), and Brepl was related to NH$_3$-N ($R^2$ = 0.104, $p < 0.05$), while there was no significant relationship between Brich and NH$_3$-N ($p > 0.05$) (Figure 5).

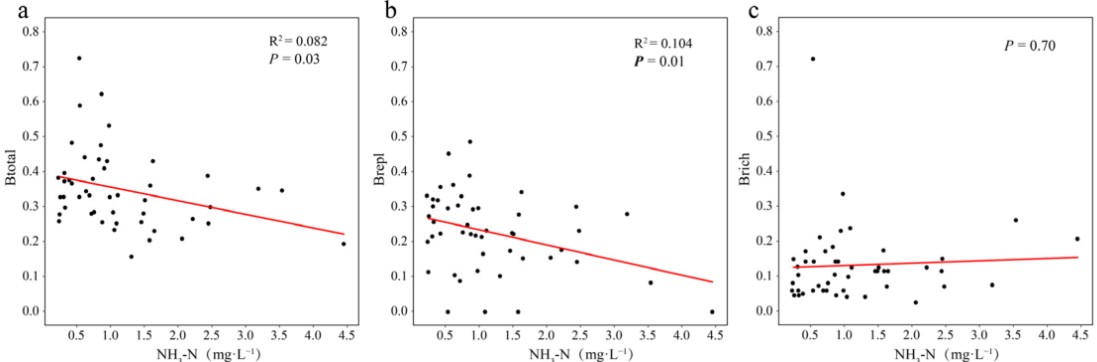

**Figure 5.** Linear regression relationships between beta diversity ((**a**): Btotal; (**b**): Brepl; (**c**): Brich) and the concentration of NH$_3$-N in the rivers of Shanghai City.

### 3.4. Environmental Factor Variations in the Nutrient Addition Experiment

The environmental variables for all the treatments are shown in Table S4. At the start of the test, the nitrogen concentrations (TN and NH$_3$-N) increased to varying degrees in the treatments after the addition of NH$_4$Cl. Compared to the initial concentrations, the contents of NH$_3$-N and TN had decreased at the end of experiment (Day 10) in all treatments, while NO$_3{}^-$-N and NO$_2{}^-$-N had increased. The concentration of Chla increased with the addition of NH$_4$Cl, except for the case of treatment B. No change was found in treatment A (no addition of NH$_4$Cl) either.

### 3.5. Effect of Ammonia Nitrogen on the Structure and Beta Diversity of Zooplankton

The initial zooplankton density was $1071 \pm 115$ ind. L$^{-1}$. At the end of the test, there was a substantial change in zooplankton density among the treatments, particularly in A (no addition of NH$_4$Cl), in which the total density decreased by 56% to $471 \pm 107$ ind. L$^{-1}$. The total density of zooplankton in treatments B, C, D, and F was similar to the initial density, while the density in the other treatments (E and G) had increased (Figure S3).

The PcoA results demonstrated significant variation in the zooplankton community structure between the treatments ($p < 0.05$) (Figure 6). PcoA1 and PcoA2 explained a total of 60.4% of the zooplankton community changes. The center coordinates of groups A, B, C, D, E, F, and G were ($-0.28$, $-0.20$), ($-0.29$, $-0.04$), ($-0.15$, 0.19), ($-0.01$, 0.09), (0.16, 0.17), (0.27, $-0.01$), and (0.30, $-0.20$), respectively (Figure 6a). The results of the PERMANOVA also demonstrated striking differences in the structure of the zooplankton assemblages, with similar dominant species in group A and groups B, C, E, F, and G ($p < 0.05$) (Figure 6b). Shared species in the different groups were the Rotifera *Polyarthra dolichoptera*, *Brachionus angularis*, and *B. calyciflorus*, the Cladocera *Daphnia carinata*, *Simocephalus vetulus*, and *Bosmina coregoni*, and the Copepoda *Thermocyclops taihokuensis*, *T. vermifer*, *T. hyalinus*, and *Mesocyclops leuckarti* (Table S5). The relative abundances of these taxa differed, with Rotifera increasing with nutrient addition and Cladocera and Copepoda decreasing.

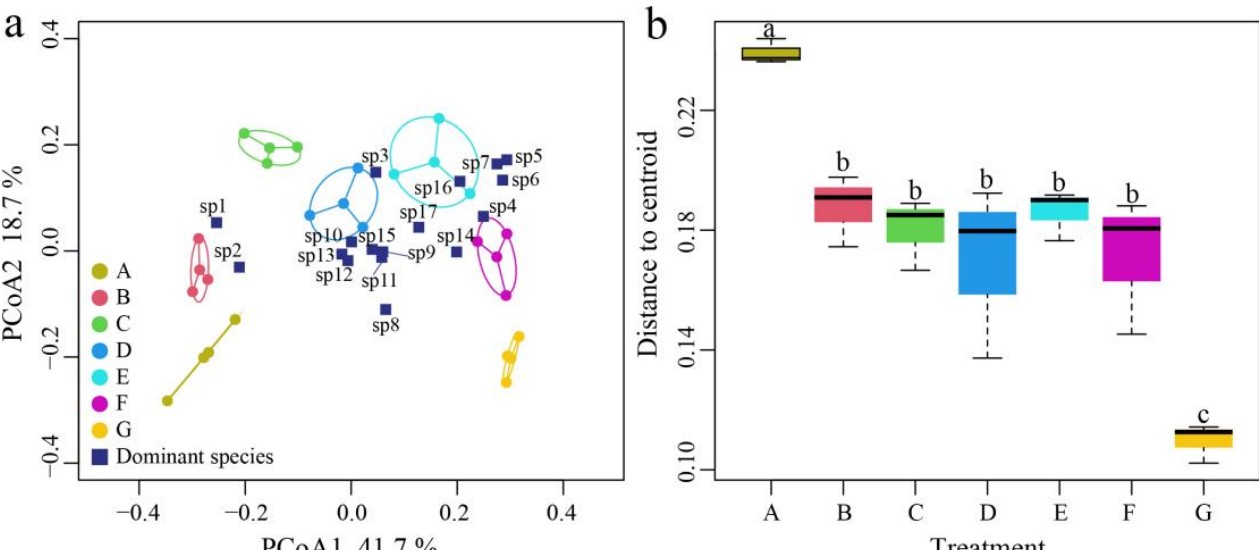

**Figure 6.** PcoA showed the differences in zooplankton community structure (**a**) and average distance to the centroid (**b**) in the treatments of the nutrient addition experiment. Different letters represent significant differences ($p < 0.05$). sp1, *Trichocerca stylata*; sp2, *Colurella obtusa*; sp3, *Polyarthra dolichoptera*; sp4, *Brachionus falcatus*; sp5, *B. urceus*; sp6, *B. forficula*; sp7, *B. angularis*; sp8, *B. calyciflorus*; sp9, *Thermocyclops taihokuensis*; sp10, *T. vermifer*; sp11, *T. hyalinus*; sp12, *Mesocyclops leuckarti*; sp13, *Daphnia carinata*; sp14, *Scapholeberis aurita*; sp15, *Simocephalus Vetulus*; sp16, *Bosmina longirostris*; sp17, *B. coregoni*.

We found major changes in zooplankton beta diversity (Figure S4) that declined with increasing $NH_3$-N concentrations. Brepl made a high contribution to the Btotal in all treatments, accounting for 56.0%, 76.9%, 79.4%, 83.7%, 74.3%, 71.2%, and 87.4% in groups A, B, C, D, E, F, and G, respectively.

## 4. Discussion

We found that the community composition of zooplankton in the investigated urban river systems was particularly affected by variations in ammonia nitrogen concentrations. Similar results have also been corroborated by other studies showing that ammonia, one of the N sources in water bodies, has a notable influence on the community structure of zooplankton [41–44]. For example, the effect of ammonia nitrogen on zooplankton community structure in Taihu Lake was more significant than the effects of other environmental factors [44]. Ammonia might facilitate the increase of Chla content and especially the growth of phytoplankton [45]. As demonstrated in our nutrient addition experiment, ammonia may stimulate phytoplankton production, and in the field survey we also found higher concentrations of Chla in Group II, which also had higher concentrations of ammonia nitrogen ($\geq 1.03$ mg $N \cdot L^{-1}$) than Group I ($< 1.03$ mg $N \cdot L^{-1}$). Higher phytoplankton abundance may lead to a higher biomass of zooplankton, as seen in our study and many other freshwater ecosystem studies [46,47]. A previous survey indicated that the phytoplankton communities in Shanghai rivers were mainly dominated by Chlorophyta, Cyanobacteria, and Bacillariophyta [48]; nutrient enrichment may alter the phytoplankton community structure towards the dominance of species such as *Aphanocapsa* spp. and *Radiocystis* sp., which are often difficult to ingest [49], altering the zooplankton community. In addition, fish, macroinvertebrate predators, competition, and macrophytes are also important factors affecting the community composition of zooplankton [46,50,51]. For example, predation by fish could directly effect variations in zooplankton community structure [52]. Another finding was that Rotifera contributed a large proportion of both Group I and Group II, which is consistent with the results from studies of other aquatic ecosystems [53,54]. The main reason is the rotifers' small body size, weak mobility, fertility, short generation times, and fast growth rates [55,56]. We found that Group I had a larger proportion of crustaceans than Group II and that the dominant and indicator species in the two groups differed. Different zooplankton species respond differently to the content of ammonia nitrogen [42]; *Diaphanosoma*, which were found in Group I, have shown better survival at low contents of ammonia nitrogen [42], while *Keratella cochlearis*, which only occurred in Group II, is an indicator species of elevated ammonia nitrogen concentrations [57]. We further found that the dominant species in both groups were indicator species of eutrophic water bodies [58], emphasizing that the urban river ecosystem of Shanghai is nutrient-loaded and eutrophic.

The total beta diversity in Group I was considerably greater than that in Group II. More favorable environmental conditions allow the coexistence of a greater proportion of species, creating an increase in species richness [59], whereas beta diversity and richness decrease under intensified environmental stress [59]. Therefore, it was suggested that the nutrient load in this river ecosystem will lead to environmental homogeneity and biotic homogeneity among zooplankton assemblages, reducing the beta diversity. This result was similar to the findings of Fu (2019) in which the total beta diversity of macrophyte communities in the studied lake decreased at high TP, and the species compositions of communities tended to be more similar [60]. Multiple regression analysis showed that there was a correlation between the nitrogen content and the beta diversity of zooplankton in the river ecosystem. In addition to the negative correlation between ammonia nitrogen and Btotal (total beta diversity), Brepl (turnover) also showed such a correlation at the level of $p < 0.05$. Conversely, Brich (nestedness) had no relevance to ammonia nitrogen. Our nutrient addition experiments also demonstrated that Btotal and Brepl decreased with the addition of ammonia nitrogen. This indicated that environmental factors have a marked impact on the beta diversity of zooplankton, particularly ammonia nitrogen. Numerous research studies indicate that beta diversity is influenced by environmental

variables [12,61,62]. For example, Souza et al. (2021) discovered that total dissolved solids and Chla explained 47% of the variation of turnover, and water temperature, Chla, total nitrogen, and inorganic carbon explained almost 60% of the variance in the nestedness of zooplankton assemblages in a river basin ($p < 0.05$). We found a connection between the concentration of nitrogen (total nitrogen and ammonia) and the beta diversity of zooplankton and a negative correlation between ammonia nitrogen and both Btotal (total beta diversity) and Brepl (turnover). Our nutrient addition experiments also demonstrated that Btotal and Brepl decreased with increasing ammonia nitrogen, indicating losses in the beta diversity of zooplankton assemblages with increasing trophic state.

We noticed that the dominance of Brich was greater in Group I than in Group II, which perhaps reflects the fact that the Group I rivers were sampled in the dry season (mainly in April), and therefore displayed reduced connectivity and dispersal between habitats [63]. The beta diversity of zooplankton exhibits a nestedness pattern (Brich) under this circumstance [64]. In contrast, increased connectivity and dispersal rates within the river system during the wet season (Group II) [65,66] emphasized the significance of the turnover pattern (Brepl) [64]. Overall, Brepl contributed most to Btotal in both Group I and Group II, suggesting that the proportion of turnover components made a stronger contribution to total beta diversity. This means that the replacement of zooplankton community structure occurred at a given site rather than because of an overall loss or gain of species. In various other freshwater environments, other researchers have also concluded that turnover components play a significant role for the total beta diversity pattern of the zooplankton community structure [25,67]. Moreover, turnover components have been essential for explaining the beta diversity of other aquatic assemblages such as fish, phytoplankton, and macroinvertebrates [68–70], and it has been shown that the turnover pattern of zooplankton assemblages in the aquatic environment is particularly sensitive to environmental factors (total phosphorus, dissolved oxygen, nitrate, and ammonia nitrogen) [53]. We found a significant relationship between Brepl and environmental factors (total nitrogen and ammonia nitrogen), suggesting that the environmental heterogeneity and trophic condition of the river contributed importantly to the beta diversity pattern of the zooplankton. Understanding the role of turnover and nestedness for beta diversity is important as it may affect conservation strategies. When the nestedness components are the main contributors to total beta diversity, regions with a high species richness requires more conservation. In contrast, when the turnover component is important, as in our study, all sites in the region contribute similarly to beta diversity; thus, all sites need to be preserved [2,71].

### 5. Conclusions

The study showed that ammonia nitrogen is a key driver of the zooplankton assemblages in the urban river ecosystem of Shanghai. The beta diversity of zooplankton assemblages was driven primarily by turnover (Brepl), suggesting that the replacement of species was dominant in the impacted rivers in this region. Moreover, multiple regression analysis revealed a negative relationship between environmental parameters (particularly ammonia) and the beta diversity of zooplankton assemblages. Our research contributes to augmenting our understanding of how environmental factors influence beta diversity patterns in the zooplankton community structures of urban river ecosystems.

**Supplementary Materials:** The following supporting information can be downloaded at: https://www.mdpi.com/article/10.3390/w15081449/s1, Figure S1: Conceptual diagram (a) and experimental simulation device (b) of nutrient addition experiment; Figure S2: Characteristics of environmental factors in the different groups; Figure S3: Zooplankton density and relative abundance in the nutrient addition experiment; Figure S4: Beta diversity of zooplankton assemblages in the nutrient addition experiment; Table S1: Sampling sites in the rivers of Shanghai City; Table S2: The relative abundances of the dominant zooplankton taxa and Beta diversity in Group I and Group II; Table S3: Linear regression relationship between beta diversity and environmental factors in the rivers of Shanghai

City; Table S4: Variations of environmental factors in the nutrient addition experiment; Table S5. The relative abundance of dominant zooplankton taxa in the different treatments.

**Author Contributions:** Conceptualization, F.Z. and L.W.; Methodology, L.W.; Software, G.S.; Formal analysis, G.S.; Investigation, C.D.; Writing—Original draft, C.D.; Writing—Review & editing, F.Z., E.J., L.Z. and W.Z.; Supervision, L.Z., W.Z. and X.F.; Project administration, L.Z. and W.Z. All authors have read and agreed to the published version of the manuscript.

**Funding:** The research was jointly funded by the Yellow River Basin Ecological Protection and High-quality Development Joint Study (Phase I) (grant number 2022-YRUC-01-0202), the National Key Research and Development Program of China (grant number 2022YFC2601305), and the Natural Science Foundation of China (grant number 41901119). E.J. was funded by the TÜBITAK program BIDEB2322 (project 118C250).

**Institutional Review Board Statement:** Not applicable.

**Informed Consent Statement:** Not applicable.

**Data Availability Statement:** All data generated or analyzed during this study are included in this published article and the supplementary information files.

**Conflicts of Interest:** The authors state that they have no conflict of interest.

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
