# Peer review of "Ammonia Influences the Zooplankton Assemblage and Beta Diversity Patterns in Complicated Urban River Ecosystems"

_water, doi:10.3390/w15081449_

Round 1

Reviewer 1 Report

Review for the paper "Ammonia influences the zooplankton assemblage and beta diversity patterns in sophisticated urban river ecosystems" by Cai-li Du, Feng-bin Zhao, Guang-xia Shang, Li-qing Wang, Erik Jeppesen, Lie-yu Zhang, Wei Zhang, Xin Fang submitted to "Water".

General comment.

River waters are ecosystems of significant human and ecological interest where complex processes occur. The interaction of physical, chemical, and ecological processes produces high spatial variability in the water column. This variability influences the abundance and structure of different biological communities present in this environment, especially phyto- and zooplankton as food chain lower levels. Zooplankton density and distribution is often related to predator/prey interactions, as a prey for fish and as consumers of phytoplankton. Therefore, zooplankton is controlled by zooplankton and fish predation, and also by nutrient availability. The present study took place at various rivers in Shanghai City, China and the authors aimed to assess the effect of environmental variables on the community structure and beta diversity patterns of the zooplankton. They revealed high nutrient (ammonium) concentration to be a factor reducing zooplankton beta-diversity. The study may be useful for monitoring of aquatic ecosystems in urban areas and expand our knowledge regarding the structure of zooplankton assemblages under anthropogenic influence. Methods are adequate and statistical treatment is comprehensive and valid. The main results are visualized in a good way. The discussion part provides a rather good interpretation of the results but may be somewhat improved. After minor revisions this paper may be accepted for publication in "Water".

Specific remarks.

L3, 82, 91. Consider replacing "sophisticated" with "complicated".

L62. Consider replacing "Zooplankton" with " Zooplankton community".

L92. Consider replacing "new light on" with "new insight into".

Fig. 1. Resolutions of the maps must be improved.

Section 2.1. must be supplemented with a short description of the environmental conditions in the region (climate, drainage basin etc,).

L131. Delete 'with a microscope'.

Results and discussion. Table S3 shows correlations between environmental variables and beta diversity of the zooplankton. However, I have not found results indicating the correlation between water temperature and zooplankton diversity. Temperature is considered to be an important predictor of the total zooplankton abundance and its effect must be studied and discussed in the ms.

Discussion.

-Compare your results with other ecosystems.

-I also recommend considering other important factors that might be responsible for the zooplankton diversity pattern in the rivers. In particular, predator pressure and biotic interactions between zooplankton groups must be discussed as possible drivers.

Reviewer 2 Report

Dera Authors,

Please find the comments in the annotated PDF as stated.

My concern is that seasonality is one of the important factors in zooplankton assemblage patterns in such a dynamic riverine environment. Authors performed one time sampling from each river systems for this study. How the authors mitigate these zooplankton seasonal dynamics by following a single sampling from the river system.

The discussion about individual rivers rather than about the groups is missing. Although statistically followed, the information about the system individual would be more helpful in making the zooplankton dynamics from the rivers more readable. 

It requires a minor revision.

The statistical methods followed for the study and interpretation is appreciable.

Good luck and stay healthy.
